# Relationship between extinction magnitude and climate change during major marine/terrestrial animal crises

Kunio Kaiho

Department of Earth Science, Tohoku University, Aoba-aza, Aramaki, Aoba-ku, Sendai 980-8578, Japan

*Correspondence to*: Kunio Kaiho (kunio.kaiho.a6@tohoku.ac.jp)

**Abstract.** Major mass extinctions in the Phanerozoic Eon occurred during abrupt global climate changes accompanied by environmental destruction driven by large volcanic eruptions and projectile impacts. Relationships between land temperature anomalies and terrestrial animal extinctions as well as the difference in response between marine and terrestrial animals to abrupt climate changes in the Phanerozoic have not been quantitatively evaluated. My analyses show that the magnitude of
major extinctions in marine invertebrates and that of terrestrial tetrapods correlate well with the coincidental anomaly of global and habitat surface temperatures during biotic crises, respectively, regardless of the difference between warming and cooling (correlation coefficient $R$ = 0.92–0.95). The loss of more than 35 % of marine genera and 60 % of marine species loss corresponding to major mass extinctions so called "big five" correlate with a > 7 °C global cooling and a 7–9 °C global warming for marine animals, and a > 7 °C global cooling and a > ~7 °C global warming for terrestrial tetrapods,
accompanied with ± 1 °C error in the temperature anomalies as the global average, although number of terrestrial data is small. These relationships indicate that (i) abrupt changes in climate and environment associated with high energy input by volcanism and impact relate to the magnitude of mass extinctions and (ii) the Anthropogenic future extinction magnitude will not reach the major mass extinction magnitude, when the extinction magnitude parallelly changes with global surface temperature anomaly. In the linear relationship, I found lower tolerance of terrestrial tetrapods than that of marine animals
for the same global warming events and a higher sensitivity of marine animals to the same habitat temperature change than terrestrial animals. These phenomena fit to the ongoing extinctions.

## 1 Introduction

There are two habitat realms for animals: marine and terrestrial realms. Major mass extinctions of animals have occurred
five times: 444, 372, 252, 201, and 66 million years ago (Ma) after fundamental animal diversification was finished at ~520 Ma, commonly marked by high extinction percentages of animals inhabiting the marine realm (Sepkoski, 1996; Bambach, 2006; Stanley, 2016; Fan et al., 2020); these events were driven by large volcanic eruptions and projectile impacts (Schulte et al., 2010; Davies et al., 2017; Burgess et al., 2017; Bond and Grasby, 2020; Kaiho et al., 2016, 2021a, 2021b, 2022). The last three mass extinctions after the initial diversification of tetrapods at ~300 Ma had high extinction percentages for
terrestrial tetrapods (Sahney et al., 2010; Benton et al., 2013) and marine animals (Sepkoski, 1996; Bambach, 2006; Stanley, 2016; Fan et al., 2020). These major biotic crises were related to abrupt global climate changes (Balter et al., 2008; Korte et

al., 2009; Finnegan et al., 2011; Chen et al., 2011; Vellekoop et al., 2014; Chen et al., 2016; Kaiho et al., 2016, 2022; Black et al., 2017) and the accompanying environmental changes, such as acid rain, ozone depletion, reduced sunlight and oceanic anoxia, driven by large volcanic eruptions and projectile impacts (Schulte et al., 2010; Bond and Grasby, 2020), and the

relationship for terrestrial and marine animals has not been quantitatively studied.

Recently, Song et al. (2021) showed that a good relationship ($R = 0.63$) between temperature change and marine extinction magnitude under uniform time intervals (averaging ~10 Myr) spanning the late Ordovician (~450 Ma) to the early Miocene (~15 Ma). However, the coincidence of temperature change and extinction magnitude is unclear. Long-term surface temperature changes did not cause mass extinctions because animals migrate to survive (McPherson et al., 2022). Abrupt

high energy input by volcanism and impact to the surface of the Earth caused abrupt climate changes accompanied by abrupt environmental destruction, leading to animal crises. I used only data sets of coincidental abrupt climate changes for biotic crises. I analyzed on the five major mass extinctions as well as the late Guadalupian crisis, considered a major mass extinction in some literature (Stanley and Yang, 1994; Rampino and Shen, 2019). The other minor crises are omitted because they remain to be studied in detail, especially the coincidence between biotic crises and climate changes.

On the modern Earth, an ongoing species extinction occurred mainly on land rather than the sea (Barnosky et al., 2011). A study on thermal tolerance of modern animals shows a higher sensitivity of marine animals to warming than terrestrial animals (Pinsky et al., 2019). However, whether this relationship holds true for ancient animals has not yet clarified. I aimed to clarify the relationship between the magnitude of biotic crises in not only marine invertebrates but also terrestrial vertebrates (tetrapods) and the global and habitat [marine or terrestrial realm] surface temperature anomalies using only

biotic crises coinciding with abrupt climate changes, to access similarity and difference in response of terrestrial and marine animals for global and habitat (land and sea) temperature anomalies and coincidental environmental changes.

Song et al. (2021) claimed that a temperature increase of 5.2 °C above the pre-industrial level at present rates of increase would likely result in mass extinction comparable to that of the major Phanerozoic events, regardless of other, non-climatic anthropogenic changes that negatively affect animal life. The 5.2 °C is not a global surface temperature anomaly but

a sea surface temperature (SST) anomaly. The global surface temperature anomaly is much higher than 5.2 °C. Fig. 1d shows the conversion between the global surface temperature anomaly, land-surface temperature anomaly (global mean), and SST anomaly (global mean) to access global and habitat (land and sea) temperature anomalies in each biotic crisis. I reached different conclusions on the surface temperature anomaly and the prediction for the future extinction magnitude for the conclusions of Song et al. (2021).


## 2 Data and methods

### 2.1 Diversity reduction percentage

Although Song et al. (2021) analyzed extinction data compared to sea surface temperature (SST) changes, there is no confirmation of the exact coincidence between extinction rate and temperature change for minor extinctions. I used only data

showing the coincidence of shallow marine extinctions and temperature changes from the same outcrop of sedimentary rocks

for a more accurate result. Therefore, I analyzed the six mass extinctions and the modern extinction, which coincided with global climate changes. For the five major mass extinctions, I used three marine genera loss percentage data sets based on three different methods (well-preserved genera data of Sepkosky, 1996; Bambach, 2006; Stanley, 2016). They show that the largest loss percentage occurred at the end of the Permian (58–66%), the smallest loss at the Frasnian–Famennian boundary

(F–F; 18–41%), and intermediate loss percentage at the other three mass extinctions (39–52 %) (Fig. 1a). The error for genera and species loss percentages is approximately ± 5 % for >15 % loss values (Stanley, 2016). I do not use their data for the end of the Guadalupian because the uncertain high loss % likely due to "smear back" (Signor-Lipps Effect) from the great end-Permian event is enhanced by the loss of record from lower sea level in the later Permian (Bambach, 2006) (Fig. 1a, Table 1). Instead, I use the marine species and genera diversity data of Fan et al. (data from China; Fan et al., 2020) for

the end of the Guadalupian, which seems to be the most believable data because their sedimentary rock sequences of the GSSP section and nearby sections contain continuous sedimentary rocks without a time gap (Fan et al., 2020; Huang et al., 2019). The data from China are likely not affected by the Signor-Lipps Effect. I also used data of % extinction of Barnosky et al. (2011) and Ceballos et al. (2015) for the Holocene–Anthropocene marine and terrestrial species.

I calculated % extinction using the marine animal diversity data of Fan et al. (2020) for the end-Guadalupian extinction

and terrestrial tetrapod diversity data of Benton (2013) and Sahney and Benton (2017) for the last four crises since the early diversification of terrestrial tetrapods in the Carboniferous. The genera (species) loss % is calculated using the formula of total number of extinction genera (species) for a mass extinction interval / total number of genera (species) in a substage just before the extinction (conventional method in Stanley [2016]).

Tetrapod genera loss of Benton (2013) and Sahney and Benton (2017) are used to represent reduction percentage for

terrestrial animals because it is difficult to obtain good data for diversity losses among insects and plants. The tetrapod species data were converted from genus extinction percentage to species extinction percentage using the relationship curve between family and genera for tetrapods in Fig. 1c since the actual marine family/genus data mostly fit the conversion relationship curve of genus/species of Stanley (2016) (Figs. 1b, c).

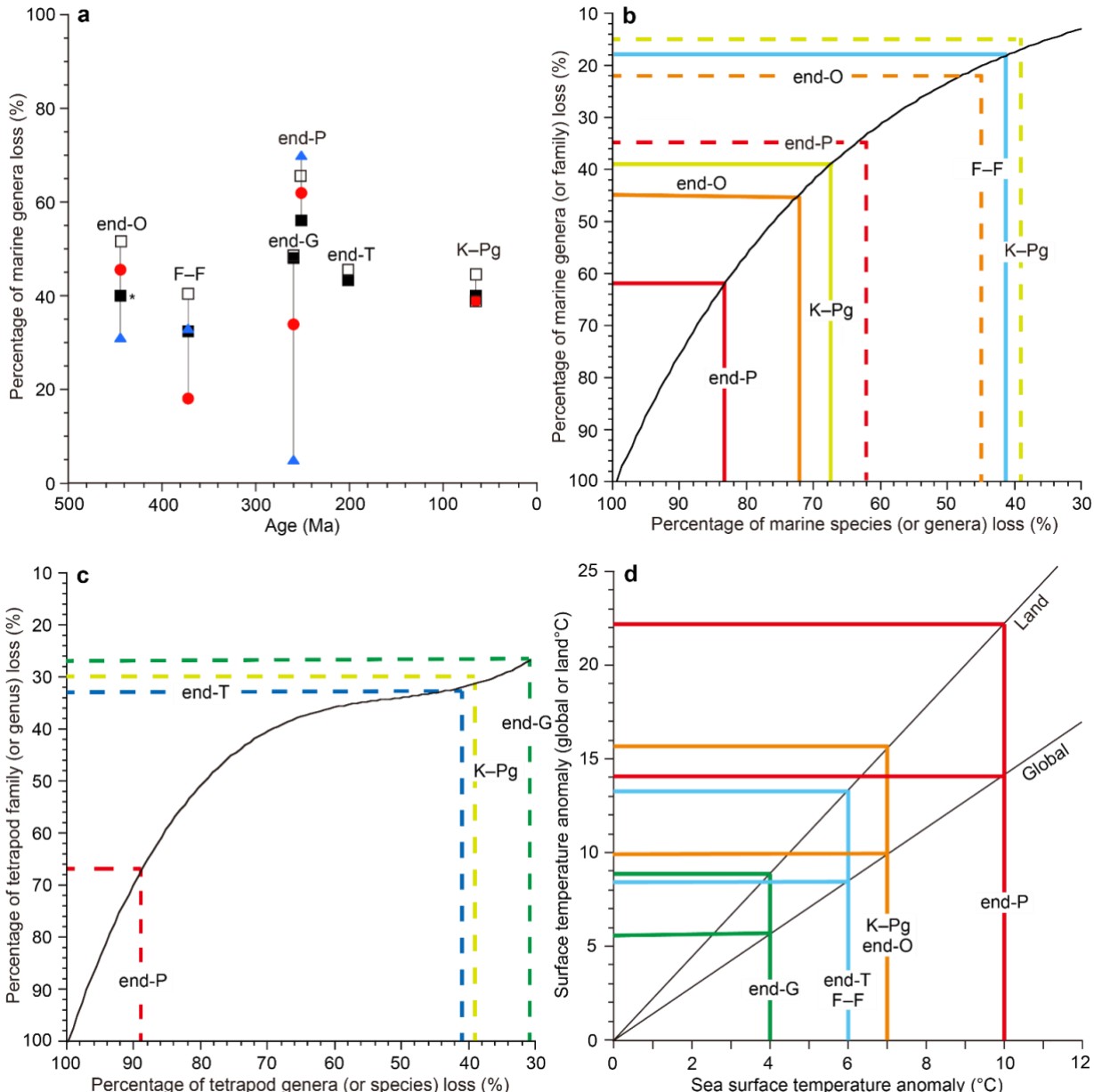

**Figure 1:** Marine genus loss (%) distribution (**a**), relationship between extinction percentages of species, genera, and families (**b** for marine invertebrates, **c** for terrestrial tetrapods) and between global surface temperature anomalies, land-surface temperature anomalies, and sea-surface temperature (SST) anomalies (**d**). Open square data from Sepkoski (1996), black square data from Bambach (2006), red circle data from Stanley (2016), and blue triangle data from Fan et al. (2020) in the graph (**a**). Graphs (**b**, **c**) were used to convert extinction percentages among species, genera, and families. Dashed lines show the genus-family loss relationship (**b**, **c**). The black curve in (**b**) is after Stanley (2016). Graph (**d**) is based on the model calculation data from Kaiho and Oshima (2017) and is used to convert between the global surface temperature anomaly, land-

surface temperature anomaly (global mean), and SST anomaly (global mean). All data are from Table 1. O: Ordovician. F–F: Frasnian–Famennian boundary. P: Permian. T: Triassic. K–Pg: Cretaceous–Paleogene boundary. *Largest extinction % in the Late Ordovician mass extinction (LOME).


**Table 1:** Marine animal and tetrapod family and genus extinction percentages and global, sea, and land-surface temperature anomalies

| Crisis | Age (Ma) | Marine family Ext (%) | Marine genus Ext (%) | Marine genus Ext (%) | Marine genus Ext (%) | Marine genus Ext (%) | Tetrapod family Ext (%) | Tetrapod Genus Ext (%) | Temp. anomary (Global °C) | Temp. anomary (SST °C) | Temp. anomary (Land °C) |
|---|---|---|---|---|---|---|---|---|---|---|---|
| H–A | 0 | - | | | | 0* | - | *0.6* | *1* | *0.7* | *1.5* |
| K–Pg | 66 | 15 | **45** | **40** | **39** | | 30 | **39*** | *-10* | *-7* | *-16* |
| End-T | 201.4 | 13 | **46** | **43** | | | 33 | **41** | *-8* | *-6* | *-13* |
| End-P | 251.9 | 35 | **66** | **56** | **62** | 70 | 67 | **89** | *14* | *10* | *22* |
| End-G | 259.8 | - | 48 | 48 | 34 | **5*** | 27 | **31** | *6* | *4* | *9* |
| F–F | 372 | - | **41** | **35** | **18** | 33 | - | - | *-7* | *-5* | *-11* |
| End-O | 445–444 | 22* | **52** | **40** | **45.5** | 31 | - | - | *10* | *7* | *16* |
| Reference | | 1, 2, 3* | 1 | 4 | 5 | 6, 8* | 9 | 10, 11* | 13, 14 | 15–20 | |

Ext (%): extinction percentage. Data marked by bold letters are used in Figure 3. Italic letters show values converted using Figure 1b–d. O: Ordovician. F–F: Frasnian–Famennian boundary. G: Guadalupian. P: Permian. T: Triassic. K–Pg: Cretaceous–Paleogene boundary. *: corresponds to a reference marked by *. **: data without brachiopods because their diversity increased spanning the G–L boundary. **39*** at K–Pg is calculated from the data. References: 1: Sepkoski (1996); 2: Rampino et al. (2020); 3: Sepkoski (1982); 4: Bambach (2006); 5: Stanley (2016); 6: Fan et al. (2020); 7: Barnosky et al. (2011); 8: Ceballos et al. (2015); 9: Sahney et al. (2010); 10: Benton et al. (2013); 11: Sahney and Benton (2017); 12: IUCN (2021); 13: Waters et al. (2016); 14: Working Group I Contribution to the Fifth Assessment Report of the Intergovernmental Panel on Climate Change (2013); 15: Vellekoop et al. (2014); 16: Korte et al. (2009); 17: Chen et al. (2016);  18: Chen et al. (2011);  19: Balter et al. (2008);  20: Finnegan et al. (2011). Italic values are converted using Figs. 1b and 1d.

## 2.2 Surface temperature anomaly

The largest absolute sea-surface temperature (SST) anomalies during each crisis were obtained from the oxygen isotope ratios ($^{18}O/^{16}O$) of marine animal fossils (Balter et al., 2008; Korte et al., 2009; Finnegan et al., 2011, Chen et al., 2011) and the organic biomarker index ($TEX_{86}$) (Vellekoop et al., 2014) (Table 1). All the SST data are from low latitudes (Table 2). Global surface temperature anomalies at low latitudes are always intermediate values (near average values) regardless of (i) source latitudes of greenhouse gases or aerosols blocking sunlight (Kaiho and Oshima, 2017) (Table 1) and (ii) global

warming and cooling because the highest anomaly appears at middle–high latitudes in the source hemisphere and the lowest anomaly appears at middle–high latitudes in the other hemisphere based on warming case data (Pinsky et al., 2019) and cooling case data (Kaiho et al., 2016). Therefore, I use each SST anomaly at low latitudes as an intermediate value (near average) in the Earth at each age. The error for the SST anomaly in geologic ages is approximately ± 1 °C including approximately ± 0.5 °C depending on the sample location to obtain the average value and approximately ± 0.5 °C depending

on detection of the largest anomaly for abrupt short-term events from sedimentary rocks, which usually deposited 1–100 mm/kyr, except for impact ejecta sediment. I converted SST anomalies of various geologic ages to global surface temperature anomalies and land-surface temperature anomalies using Fig. 1d, which was generated from global cooling and warming (recovery) data of the climate model calculation (Kaiho and Oshima, 2017) (Fig. 1d).

## 2.3 Relationships between taxa loss % and temperature anomaly

Finally, I show Pearson's correlation coefficient $R$ between taxa loss % and absolute habitat temperature (sea surface for marine faunas and land surface for terrestrial faunas) for the three marine genera loss percentage data sets and a terrestrial data set. To calculate their correlation coefficient, marine end-G and H–A data are included into each data set, because the small percentage values cannot be changed largely due to the method variation (Table 3).

**Table 2:** Source latitudes of causal gas and aerosols and SST data

| Crisis | Source of causal gas and aerosols | SST data site |
|--------|-----------------------------------|---------------|
| K–Pg | ~25°N | ~30°N |
| end-T | ~20°S – ~30°N | ~30°N |
| end-P | ~50°N | ~15°N |
| end-G | ~30°N | ~30°N |
| F–F | ~10°S – ~30°N | ~25°S |
| end-O | ? | ~20°S – ~10°S |

O: Ordovician. F–F: Frasnian–Famennian boundary. G: Guadalupian. P: Permian. T: Triassic. K–Pg: Cretaceous–Paleogene boundary.

**Table 3:** Marine and terrestrial genera and species extinction %, absolute SST anomaly and land temperature anomaly, and their Pearson's correlation coefficient *R* for Figure 3

| Crisis | Marine Sepkosky M genus Ext (%) | Marine Sepkosky M species Ext (%) | Marine Bambach M genus Ext (%) | Marine Bambach M species Ext (%) | Marine Stanley M genus Ext (%) | Marine Stanley M species Ext (%) | Absolute SST anomaly (°C) | Tetrapod Genus genus Ext (%) | Tetrapod species species Ext (%) | Absolute land Temperature anomary (°C) |
|---|---|---|---|---|---|---|---|---|---|---|
| H–A | 0 | 0 | 0 | 0 | 0 | 0 | *0.7* | *0.6* | *1* | *1.5* |
| K–Pg | **45** | **72** | **40** | **68** | 39 | 68 | 7 | 39 | 67 | *16* |
| End-T | **46** | **73** | **43** | **70** | | | 6 | 41 | 70 | *13* |
| End-P | **66** | **86** | **56** | **80** | 62 | 83 | 10 | 89 | 97 | *22* |
| End-G | 5 | 11 | 5 | 11 | 5 | 11 | 4 | 31 | *38* | *9* |
| F–F | **41** | **69** | **35** | **62** | 18 | *42* | 5 | - | - | *11* |
| End-O | **52** | **77** | **40** | **68** | 45.5 | 72 | 7 | - | - | *16* |
| Correlation *R* | 0.92 | 0.88 | 0.92 | 0.88 | 0.95 | 0.95 | | 0.95 | 0.98 | |

Bold roman marine taxa extinction % data are from Sepkoski (2006) in Sepkoski M column, Bambach (2006) in Bambach M (method) column, and Stanley (2016) in Stanley M column. Data marked by light letters of extinction % are from Fan et al. (2020) for End-G, Barnosky et al. (2011) and Ceballos et al. (2015) for H–A. Italic values are converted using Figure 1b–d. See Table 1 for the other explanations.

## 3 Results

### 3.1 Magnitude of marine/terrestrial crises

My analysis of the major mass extinctions shows that the Late Ordovician mass extinction (LOME) was marked by only a marine crisis (40–52 % genera loss and 68–77 % species loss in the two methods) since terrestrial tetrapods had not yet appeared. The Late Devonian mass extinction (LDME) resulted in the loss of 18–41 % genera and 42–69 % species for marine animals at the Frasnian–Famennian boundary (F–F) (I ignored the tetrapod extinction percentage due to the very low apparent diversity) (Kaiho et al., 2016). The last three major mass extinctions, end-Permian, end-Triassic, and Cretaceous–Paleogene (K–Pg) boundary, were characterized by high extinction percentages of both marine and terrestrial genera (marine: 56–66 %, 43–46 %, and 39–45 %; terrestrial: 89 %, 41 %, and 39 %; Fig. 2, Table 1) and species (marine: 80–86 %, 70–73 %, and 68–72 %; terrestrial: 97 %, 70 %, 67 %; Fig. 2, Table 1). In total, the five major mass extinctions were marked by high marine genera and species extinction percentages (18–62 % and 35–83 %). However, the end-Guadalupian extinction was marked by low marine genera and species loss (5 % and 11 %) and higher terrestrial genera and species loss (31 % and 38 %), corresponding to a major terrestrial crisis, not a major mass extinction, accompanied by a large reduction in shallow marine fusulinids (Feng et al., 2020) and reef animals (Fluegel and Kiessling, 2002) due to terrestrial disturbance. The Paleozoic biotic crises during global warming following the diversification of tetrapods had higher extinction percentages of terrestrial animals than of marine animals, but the Mesozoic biotic crises during global cooling had similar percentages of terrestrial and marine animals.

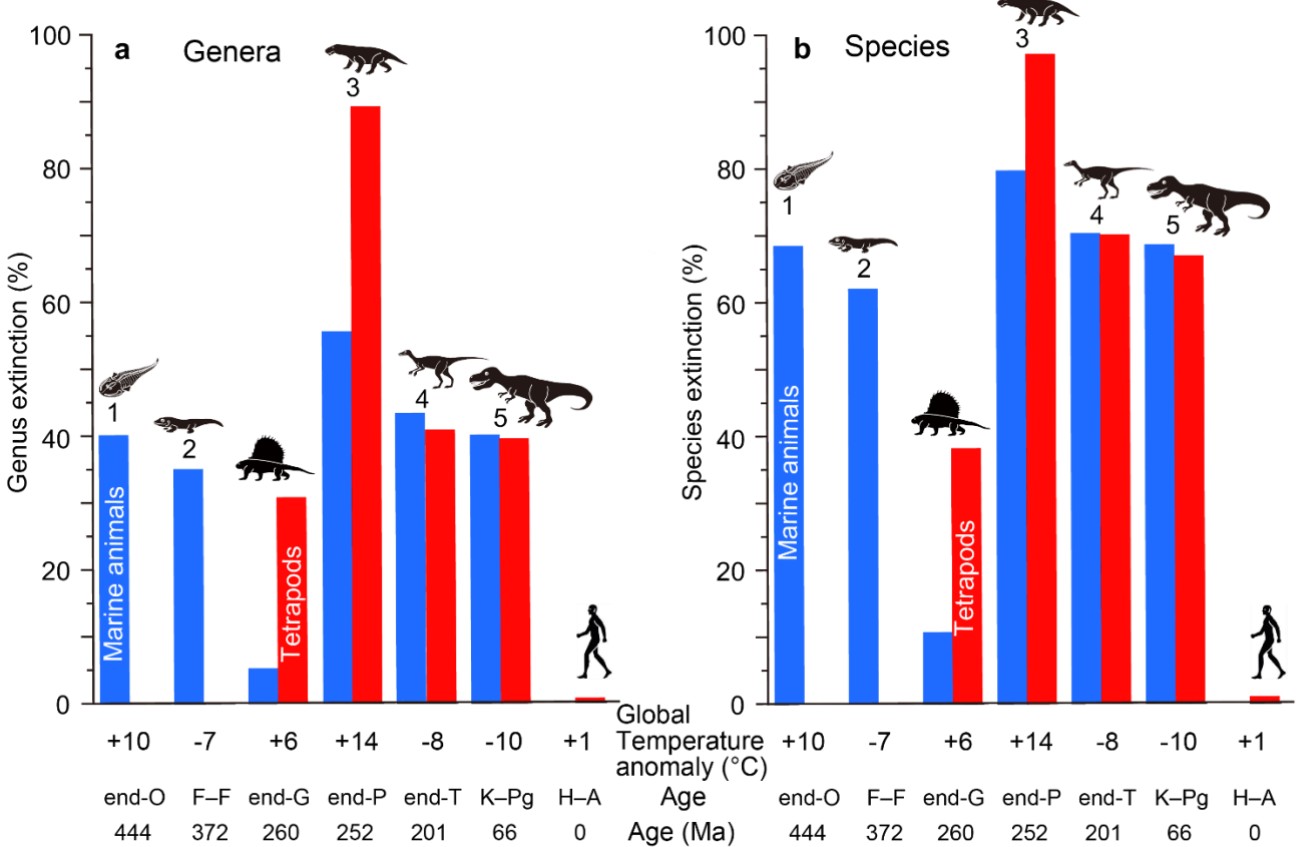

**Figure 2:** Genera (**a**) and species (**b**) extinction percentages of marine animals and tetrapods for major mass extinctions and the end-Guadalupian and Holocene–Anthropocene crises. All data are from Table 1. Marine genus and species extinction values shown by blue columns based on Bambach (2006) for genera and my calculation using Fig. 1b for species. Terrestrial genus and species extinction values shown by red columns based on Benton et al. (2013) and Sahney and Benton (2017) for genera and my calculation using Fig. 1b for species. These extinction % data are comparable because of usage of similar methods (conventional method and substage intervals). Global temperature anomaly: Global surface temperature anomaly. O: Ordovician. F–F: Frasnian–Famennian boundary. G: Guadalupian. P: Permian. T: Triassic. K–Pg: Cretaceous–Paleogene boundary. H–A: Holocene–Anthropocene (1850 to 2010 on going). Numbers 1 to 5: five major mass extinction. Each silhouette shows a representative vertebrate animal from each age.

**3.2 Sea-surface temperature anomaly during crises**

There are two extinction levels on the LOME at the Katian−Hirnantian boundary (445.2 Ma) and late Hirnantian (~444 Ma) (Bond and Grasby, 2020). Between the two extinctions, global cooling occurred, as evidenced by conodont apatite oxygen isotopes and glacial deposits (Finnegan et al., 2011); however, the two extinction levels coincided with the two shorter-term global warming events based on the oxygen isotope data of conodont apatite (Bond and Grasby, 2020). I select the maximum

anomaly +7 °C SST (~$10^5$ years) from the data of Finnegan et al. (2011) for the LOME. The trigger is estimated to be volcanism as evidenced by coincidental mercury concentration for LOME (Jones et al., 2017; Bond and Grasby, 2020).

The LDME is composed of the Frasnes, Kellwasser, and Hangenberg crises at 383, 372, and 359 Ma, respectively, and the Kellwasser is the largest crisis (Barash, 2016). The trigger was the large igneous province (LIP) emplacement of Viluy and PDD LIPs, as evidenced by mercury and coronene concentrations (Racki, 2020; Kaiho et al., 2021b). I use the largest abrupt cooling marked by a -5 °C SST anomaly (~$10^4$ years) at the Lower Kellwasser and the Upper Kellwasser crises from the oxygen isotope data of conodont apatite of Balter et al. (2008) for LDME, whereas the long-term gradual SST change between the crises shows global warming (+6 °C SST anomaly in 5 x $10^5$ years).

The Late Guadalupian crisis (LGC) occurred in the mid-Capitanian, 262 Ma, followed by the Guadalupian–Lopingian (G−L) boundary event at 259 Ma (Chen and Xu, 2019). The coincidental volcanic eruptions of the Emeishan Large Igneous Province (ELIP) in South China are thought to be the trigger of the crisis (Chen and Xu, 2019), as evidenced by mercury concentration peaks beginning in the mid-Capitanian and peaking during the G−L transition (Grasby et al., 2016). The largest abrupt anomaly +4 °C SST (~$10^5$ years) coinciding with the volcanism and extinction at the G−L boundary from the data of Chen et al. (2011) is used for LGC.

The largest biodiversity loss in the Phanerozoic occurred at the end of the Permian, with local extinction during the earliest Triassic, ~252.0–251.9 million years ago (Song et al., 2013; Kaiho et al., 2021a), marking the end of the Paleozoic. The LIP in Siberia caused sill emplacement and large eruptions at that time (Burgess et al., 2017). The coincidence of volcanic eruption and the biotic crisis was shown using the correlation of mercury, coronene, and coal fly ash (Grasby et al., 2011, 2013; Kaiho et al., 2021a). I use the largest anomaly +10 °C SST (2 x $10^4$ years) from just before the mass extinction (Bed 24) to the first minimum $\delta^{18}O$apatite value (base of Bed 27) at GSSP Meishan based on new conodont apatite $\delta^{18}O$ data showing the 2.5 permil anomaly of Chen et al. (2016) for the end-Permian mass extinction (EPME).

The age of the end-Triassic mass extinction (ETME) is estimated to be 201.564 ± 0.015 Ma, which corresponds to the emplacement of the Central Atlantic Magmatic Province (CAMP, 201.6 to 201.0 Ma) (Davies et al., 2017). The SST anomaly during the crisis is estimated as -6 °C (~$10^3$ years cooling during ~$10^4$-years) from the averaged $\delta^{18}O$ of oyster shells, assuming stable salinity, which was followed by long-term (2 x $10^5$ years) global warming (Korte et al., 2009; Kaiho et al., 2022). There were no crises during the long-term warming (Kaiho et al., 2022). I use the -6 °C anomaly for ETME, indicating global cooling.

Only the K–Pg mass extinction (KPME) at 66 Ma occurred as the result of an asteroid impact (Schulte et al., 2010). This impact produced large amounts of soot and sulfuric acid aerosols in the stratosphere by the ignition and melting of sedimentary rocks (Kaiho et al., 2016; Kaiho and Oshima, 2017). Stratospheric aerosols efficiently absorb and scatter solar radiation and reduce sunlight reaching the Earth's surface, which induces strong global cooling and a significant decrease in precipitation, particularly over equatorial areas, over ten years, with the maximum occurring in the second year (Kaiho et al., 2016; Kaiho and Oshima, 2017). Organic biomarker $TEX_{86}$ values show -7 °C as the SST largest absolute anomaly during the crisis (Vellekoop et al., 2014). This SST anomaly is consistent with the -10 °C global cooling estimated by climate model

calculations and the survival of equatorial crocodilians (Kaiho et al., 2016). The global cooling duration is ~10 years based on impact-induced climate model calculations (Kaiho et al., 2016; Kaiho and Oshima, 2017). When the Deccan Traps

volcanism also contributed to the global cooling the duration could have been longer (10–10$^2$ years) for one pulse (Schmidt, 2016).

### 3.3 Relationship between extinction magnitudes and surface temperature anomalies

I compare those data on each biotic crisis based on an assumption that the Earth and contemporary life at the time of each
220 crisis are themselves more-or-less comparable through time. My results for the relationship between past mass extinctions and surface temperature anomalies show the following features. A 4 °C SST warming was detected at the end of the Guadalupian (Chen et al., 2011), equivalent to 9 °C warming on land (6 °C global warming), as shown in Fig. 1d, which corresponded to 5% and 11% marine genera and species extinction and 31% and 38% terrestrial genera and species extinction, respectively (Figs. 3a, 3d; Table 1). The end-Ordovician mass extinction had higher temperature anomalies and
225 higher extinction percentages (40–46 % and 68–72 % marine genera and species, respectively). The EPME was marked by the highest temperature anomalies (10 °C SST [Chen et al., 2016], 22 °C on land, and 14 °C global warming) and the highest extinction percentages (56–62 % and 80–83 % marine genera and species and 89 % and 97 % terrestrial tetrapod genera and species, respectively). In contrast, the Frasnian–Famennian (F–F) boundary, the end-Triassic and Cretaceous–Paleogene (K–Pg) boundary mass extinctions coincided with the 5 °C, 6 °C, and 7 °C SST (Balter et al., 2008; Korte et al., 2009;
Vellekoop et al., 2014) cooling corresponding to 11 °C, 13 °C, and 16 °C cooling on land and 7 °C, 8 °C, and 10 °C global cooling (Fig. 1). The F–F crisis corresponds to 18–35 % and 42–62 % marine genera and species loss, respectively, that is, a marine crisis and the smallest major mass extinction, respectively. The ETME correlated with 43 % and 70 % marine genera and species loss and 41 % and 70 % terrestrial tetrapod genera and species loss, respectively, and the KPME correlated with 39–40 % and 68 % marine genera and species loss and 39 % and 67% terrestrial tetrapod genera and species loss,
respectively (Figs. 3a, d). These results indicate that a larger absolute value of the global temperature anomaly corresponds to a higher extinction percentage in the marine and terrestrial realms, regardless of whether the change is due to global warming or global cooling, considering a ± 5 % error (Figs. 3c, f). The correlation coefficient $R$ between marine extinction % and absolute SST anomaly is 0.92–0.95 for genus and 0.88– 0.95 for species, and that between terrestrial extinction % and absolute land temperature anomaly is 0.95 for genus and 0.98 for species (Figs. 3c, 3f, Table 3). There is
little or no difference to $R$ for marine animals in the three data sets. Differences in methods do not affect the conclusions. These new data indicate that global warming temperatures (> 9 °C) inducing major marine extinctions is likely higher than that of global cooling (> 7 °C). These relationships indicate that abrupt temperature anomalies and coincidental environmental changes associated with high energy input by volcanism and impact relate to the magnitude of mass extinctions.

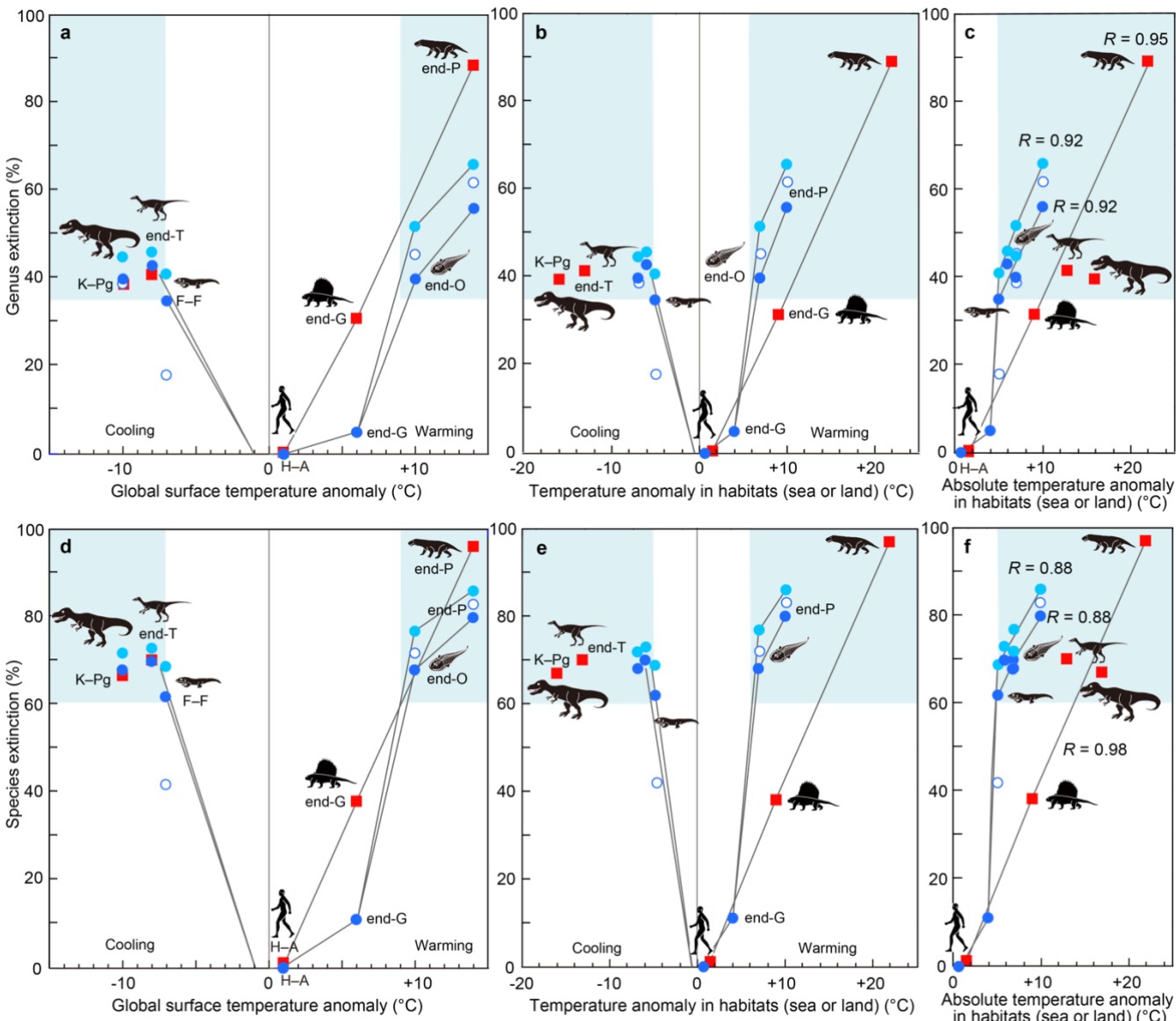

**Figure 3:** Relationship between genera and species extinction percentage and surface temperature anomaly in major mass extinctions, the end-Guadalupian crisis, and the current crisis in the Anthropocene. All vertical axes show genus or species extinction (%). (**a**)–(**c**): genera extinction. (**d**)–(**f**): species extinction. (**a**) and (**d**): relationship between that and global surface anomaly. (**b**) and (**e**): relationship between that and surface temperature anomaly in habitats (global sea or land). (**c**) and (**f**): relationship between that and absolute surface temperature anomaly in habitats (global sea or land). Blue circles: marine extinctions [light blue solid circles: Sepkoski (1996; conventional method and 107 interval data) and data of end-G and H–A; blue solid circle: Bambach (2006; conventional method and 165 substage interval data), open circle: Stanley (2016; new method and substage interval data)]. Red squares: terrestrial extinctions represented by tetrapods (calculated

from data of Benton et al. (2013; substage intervals) for end-O, F–F, end-P, end-T and Sahney et al. (2017) for K–Pg. All data are from Table 3. Comparable data sets are blue solid circles and red squares due to similar methods (conventional method and substage interval data). Pale blue areas show major extinctions. O: Ordovician. F–F: Frasnian–Famennian boundary. G: Guadalupian. P: Permian. T: Triassic. K–Pg: Cretaceous–Paleogene boundary. H–A: Holocene–Anthropocene. I also show correlation coefficient $R$ between marine extinction % and absolute SST anomaly and that between terrestrial extinction % and absolute land temperature anomaly based on the conventional method.

## 4. Discussion

### 4.1 Summary and interpretation on extinction–temperature relationship

When summarizing and interpreting these results for the past six representative crises, I find the following four news. (I) Higher global surface temperature anomalies correspond to higher extinction percentages in both marine and terrestrial realms, respectively (Figs. 3a, 3d), which suggests that climate change and related or coincidental environmental destruction is the main cause of mass extinctions on land and in sea. (II) > 35 % genera and > 60 % species loss correspond to 7–9 °C global warming and > 7 °C global cooling for marine animals, and > 7 °C global cooling and > ~7 °C global warming for terrestrial tetrapods, although number of terrestrial data is small (Figs. 3a, 3d). This relationship contains higher extinction percentages in the terrestrial realm (tetrapods) than in the marine realm (invertebrate) under the same global temperature anomaly in warming events and similar extinction percentages in the terrestrial realm (tetrapods) and the marine realm (invertebrate) in cooling events (Figs. 3a, 3d). The possible interpretations are (a) lower tolerance of terrestrial tetrapods for warming, (b) higher tolerance of marine animals for global warming, and (c) different ages and taxa-groups. I regard the lower tolerance of terrestrial tetrapods for warming as the most likely answer because major terrestrial crises occur under lower global temperature anomalies than major marine crises in global warming (Figs. 3a, 3d), which implies that major terrestrial crises occurred more frequently than major marine crises evidenced by nine decreases in tetrapod diversity having ≥ 35 % genus loss during the late Carboniferous to Early Jurassic (Benton et al., 2013) compared with two marine crises having ≥ 35 % genus extinctions during the same interval (Bambach, 2006). This is consistent with the higher extinction rate of terrestrial tetrapods compared to marine animals in the current Earth, however, the consistence is due to a different cause as anthropogenic collapse of nature, which usually parallelly occurred with global warming (e.g. Waters et al., 2016), for causes of major mass extinctions. (III) Although the ratio of the surface temperature anomaly in the terrestrial realm to that in the marine realm is 2.2 (Fig. 1d), marine animals are more likely to become extinct under a lower habitat temperature anomaly than tetrapods regardless of the difference between warming and cooling (Figs. 3c, 3f). This is possibly due to a higher sensitivity of marine animals to temperature change than terrestrial animals, which have access to places of refuge based on the current global temperature and thermal tolerance data (Pinsky et al., 2019). (IV) A similar absolute habitat temperature anomaly corresponds to a similar extinction magnitude in marine animals and terrestrial tetrapods, respectively. In other words, correlation coefficient R is very high (0.92–0.95 in marine genera and 0.95 in terrestrial genera) between

absolute habitat temperature anomaly and extinction magnitude (Figs. 3c, 3f), which indicates that the cause of the biotic crises is surface temperature anomaly and related environmental changes.

My calculation results on extinction % are comparable with the genera extinction % data in substages of Bambach and likely Sepkosky due to the usage of similar methods. Whereas, the Stanley method considered background extinction, which differs from my calculations (e.g. low extinction % for F–F of Stanley is due to the consideration of the high background extinction % [Fig. 1a]). I studied only biotic crises showing coincidence of surface temperature change and an extinction, resulting in a very high correlation coefficient (0.92–0.95) between absolute SST anomaly and extinction magnitude on marine fossils in three data sets based on the three methods of Sepkosky, Bambach, and Stanley ($R$ = 0.92, 0.92 and 0.95 for genus). The correlation coefficient of Song et al. (2021) is much lower ($R$ = 0.63 for genus), which is likely due to the low correlation in low extinction rates. It is likely due to the lack of sensitivity of marine animals for small temperature change or the usage of an uncertain coincidence with global climate changes. Song et al. (2021) concluded that a temperature increase of 5.2 °C above the pre-industrial level at the present rate of increase would likely result in a major marine mass extinction. The 5.2 °C is SST anomaly, which corresponds to a 7 °C global surface temperature anomaly (Fig. 1d). This is consistent with my results for global cooling, however, a 9 °C global surface warming is essential for a major marine mass extinction (Figs. 3a, 3d). The physical law of the temperature anomaly extinction relationship shown in Figs. 1d and 3 controls the extinction of terrestrial and marine animals, as shown in Fig. 2.

### 4.2 Climate changes and causes of mass extinctions

McPherson et al. (2022) argued that slow temperature changes will provide opportunities for species to adapt, thus, the rapidity of environmental change produced by abrupt climate change is fundamentally more important than the magnitude of the change alone for mass extinctions. The duration of global cooling and warming of the large volcanic eruptions during the five crises were $10^4$ years and $10^4$–$10^5$ years, respectively. In the case of the cooling crises, much shorter climate change events could have repeatedly occurred because one large eruption causes a ~10-year global cooling pulse (Timmreck et al., 2012). For example, a $10^4$-year cool-climate-period corresponding to the end-Triassic mass extinction likely contained numerous cooling pulses causing the 6 °C SST reduction over <$10^3$ years in total (Kaiho et al., 2022). Thus volcanic eruptions cause repeated abrupt (10 years) global cooling pulses, whereas a bolide impact would cause only one cooling pulse of ~10 years (Kaiho et al., 2016). Global warming lasts $10^4$–$10^5$ years in the case of volcanic events. Coincidental environmental changes should relate to the magnitude of mass extinctions.

The significant relationship between the surface temperature anomaly and extinction magnitude indicates that the cause of major extinctions is surface temperature change and coincidental environmental changes, such as acid rain, ozone depletion, reducing sunlight, desertification, soil erosion, and oceanic anoxia, driven by large volcanic eruptions and projectile impacts; these causal climatic and environmental conditions changed in parallel due to the same controls as each volcanism and impact. These climatic and environmental anomalies are controlled by stratospheric aerosols, such as sulfuric

acid and black carbon, for reducing sunlight – global cooling – acid rain, halogen for ozone depletion, and atmospheric greenhouse gases, such as $CO_2$ and methene, for surface warming.

Global cooling and warming have been reported in many periods in the Phanerozoic based on oxygen isotopes (Stanley, 2010); however, most of them are long-term climate changes. When surface temperature changes slowly ($>\sim10^3$ years), animals migrate and survive; an abrupt temperature change and accompanied environmental change is thought to be essential

for mass extinctions. There were no significant marine extinctions during global warming of two famous global warming events at the end-Cenomanian and Paleocene–Eocene transitions (Kaiho, 1994); which were due to volcanism under the oceanic crust (Bond and Wignall, 2014). This type of volcanism cannot eject volcanic $SO_2$ gas into the stratosphere, resulting in no short-term global cooling and gradual global warming by the gradual release of $CO_2$ from volcanism under the ocean; conversely, the Late Devonian, end-Permian, and end-Triassic LIPs were emplaced on land, resulting in $SO_2$ gas emissions

into the stratosphere, causing short-term global cooling and accompanying environmental changes, followed by longer-term global-warming due to volcanic greenhouse gas emissions. An eruption causes global cooling that lasts for a few to ten years; thus, detection is difficult; however, LIP volcanism causes thousands of eruptions (Svensen et al., 2009), resulting in the detection of decreases in SST from sedimentary rocks when the release of $SO_2$ gas to the stratosphere exceeds $>10^3$ years (Kaiho et al., 2022), but no detection occurs in cases of $< 10^2$-year $SO_2$ emissions. Global cooling is followed by global

warming due to the cessation of $SO_2$ release to the stratosphere and the accumulation of $CO_2$ in the atmosphere from volcanisms (Kaiho et al., 2022). Global warming lasts for a long time (usually $10^4$–$10^5$ years), resulting in easy detection.

Global warming has been detected in some volcanic and impact cases, whereas global cooling has been detected from (i) sedimentary rocks formed under volcanism characterized by massive $SO_2$ gas emissions and relatively low $CO_2$ emissions by low-temperature volcanism to the stratosphere (ETME) (Kaiho et al., 2022) and (ii) quickly deposited impact ejecta

(Vellekoop et al., 2014) near the impact crater in an impact case (KPME). There is a possibility of undetected short-term global cooling before global warming in the other volcanism-induced major biotic crises. Larger volcanisms generally cause larger $SO_2$, $CO_2$, and halogen emissions, which could have resulted in a significant relationship between the global warming temperature anomaly and extinction magnitude, even if the real main cause of crises is reduced sunlight – global cooling – acid rain, ozone depletion or oceanic anoxia. Therefore, the relationship between the absolute temperature anomaly and

extinction magnitude is shown in Figs. 3c and 3f. The significant relationship in marine and terrestrial animals clarified in this study indicates that the global climate and the accompanying environmental changes are related to the magnitude of mass extinctions. Although Song et al. (2021) claimed that a temperature increase of 5.2 °C above the pre-industrial level at present rates of increase would likely result in mass extinction comparable to that of the major Phanerozoic events, regardless of other, non-climatic anthropogenic changes that negatively affect animal life; the temperature increase is not 5.2

°C, but 9 °C. The 9 °C global warming will not appear in the Anthropocene at least till 2500 under the worst scenario (IPCC, 2013; IUCN 2021; Tebaldi, et al., 2021). Prediction of the Anthropogenic future extinction magnitude using only surface temperature is difficult, because the causes of the anthropogenic extinction differ from causes of mass extinctions in geologic

time. However, I can predict that the Anthropogenic future extinction magnitude will not reach the major mass extinction magnitude, when the Anthropogenic future extinction magnitude parallelly changes to global surface temperature anomaly.


## 5. Conclusions

I conclude that the relationship between extinction magnitude and climate change during major marine/terrestrial animal crises is very high. There is a significant relationship ($R = 0.92–0.95$) between extinction magnitude of marine invertebrates and absolute SST anomaly as well as that of terrestrial tetrapods and absolute land-surface temperature anomaly ($R = 0.95–$

$0.98$). The $> 35$ % genera and $> 60$ % species loss correlate to a $> 7$ °C global cooling and a $7–9$ °C global warming for marine animals, and a $> 7$ °C global cooling and a $> \sim7$ °C global warming for terrestrial tetrapods. These relationships indicate that abrupt temperature anomalies and coincidental environmental changes associated with abrupt high energy input by LIP volcanism and that an asteroid impact relates to the magnitude of mass extinctions. The Anthropogenic future extinction magnitude will not reach the major mass extinction magnitude, when the extinction magnitude parallelly changes

with global surface temperature anomaly. In the linear relationship, I found that (i) lower tolerance of terrestrial tetrapods than that of marine animals for the same global warming events and (ii) a higher sensitivity of marine animals to the same habitat temperature change than terrestrial animals, which have access to places of refuge. These phenomena will be appeared in future hundred years.

**Author contribution:** Conceptualization: KK. Methodology: KK. Investigation: KK. Visualization: KK. Writing—original draft: KK. Writing—review & editing: KK.

**Competing interests:** The authors declare that they have no competing interests.

**Acknowledgments:** This study was supported by the Japan Society for the Promotion of Science (KAKENHI—Grants in-Aid for Scientific Research; Grant Numbers #25247084 for K.K. I thank anonymous referees for useful comments.

**Data and materials availability:** All data is available in the main text.

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
