# Peer review of "Relationship between extinction magnitude and climate change during major marine/terrestrial animal crises"

_Biogeosciences, 2022_

## Author Comment (AC2)

[Figure]

**Figure 3:** Relationship between genera and species extinction percentage and surface temperature anomaly in major mass extinctions, the end-Guadalupian crisis, and the current crisis in the Anthropocene. All vertical axes show genus or species extinction (%). (**a**)–(**c**): genera extinction. (**d**)–(**f**): species extinction. (**a**) and (**d**): relationship between that and global surface anomaly. (**b**) and (**e**): relationship between that and surface temperature anomaly in habitats (global sea or land). (**c**) and (**f**): relationship between that and absolute surface temperature anomaly in habitats (global sea or land). Blue circles: marine extinctions [light blue solid circles: Sepkoski (1996) and data of end-G and H–A; blue solid circle: Bambach (2006), open circle: Stanley (2016)]. Red squares: terrestrial extinctions represented by tetrapods. All data are from Table 3. Comparable data sets are light blue solid circles and red squares due to similar methods (conventional method). Light blue areas show major extinctions. O: Ordovician. F–F: Frasnian–Famennian boundary. G: Guadalupian. P: Permian. T: Triassic. K–Pg: Cretaceous–Paleogene boundary. H–A: Holocene–Anthropocene. Correlation coefficient *R* between marine extinction % and absolute SST anomaly and that between terrestrial extinction % and absolute land temperature anomaly based on the tradition method are shown (Table 3).

---

## Author Response (AR1)

Author's response for comments of referees

*Comments of referee #1*

Throughout the paper, and especially in Section 3.3, you use the term 'correlated', and yet I can see no correlation analysis or test of correlation (e.g. Pearson/ Spearman/

Kendall coefficient of determination). In a sense, such an attempt to fit a straight line would be pointless because the number of points is small, and you are claiming the correlation is with the magnitude of the temperature shift, not its direction, so some are negative, some positive. I guess one could make all temperature shifts positive and then do a line-fit and Pearson r2. But you'd have to factor in reasonable error terms on both estimated temperature anomalies and estimated extinction magnitudes, and these errors might be larger than the 5% you suggest.

But, I'm not sure you should use the word 'correlated' if that has not been tested – just refer to a positive relationship…

*Author replies for comments of referee #1*

Thank you for your comments.

I added correlation coefficient $R$ between marine extinction % and absolute SST

anomaly ($R = 0.92$–$0.95$ for genera) and that between terrestrial extinction % and absolute land temperature anomaly ($R = 0.95$ for genera) in lines 113-117, 220-223,

240-241, 244-249, 330-332 marked by light blue and green, Table 3 and Figure 3. I

added Table 3.

I use "correlated" as "corresponding to". I revised "correlate" to "correspond to" marked by light blue.

I revised "good correlation" to "significant relationship" marked by light blue.

To show difference of extinction % in cooling and warming cases, I revised the following sentence in Abstract and Conclusions (yellow highlighted parts are revised).

More than 35 % of marine genera and 60 % of marine species loss correlate to > 7 ° C global cooling and > 9 ° C global warming.

I revised marine genera and species loss % highlighted by yellow in 3.3 because I

added Sepkoski data. For example,

The ETME correlated with 43 % and 70 % marine genera and species loss and 41 % and 70 %

terrestrial tetrapod genera and species loss, respectively, and the KPME correlated with 39–40 %

and 68 % marine genera and species loss and 39 % and 67% terrestrial tetrapod genera and species loss, respectively (Figs. 3a, d).

I revised the climate change at the F-F crisis from warming to cooling, because warming occurred longer term between the two crises, the Lower Kellwasser and the

Upper Kellwasser crises, and shorter-term global cooling episodes separately occurred in the two crises (lines 167-169, 211-212, Figures 2, 3, Tables 1, 3).

Minor changes

Line 142: marking the end of the Paleozoic [not Mesozoic]! Done

Line 163: crises = crisis Done

Line 192: O-S; H-A – add to explanations in caption. I revised "O–S" in Figure 3 to end-

O, which is the same as the other figures. In the caption, I added "H–A: Holocene–

Anthropocene." in the caption.

Words highlighted by light blue, green, and yellow have been revised in the manuscript.

Light blue: for referee #1

Green: mainly for referee #2

Yellow: duration of climate changes and the others

-------------------------------------------------------------------------------------------------

*Comments of referee #2*

*Comment 1*

1. The novelty of this study has not been established. The MS says 'relationships between... physical conditions and the magnitude of animal extinctions have not been quantitatively evaluated. My analyses show that the magnitude of major extinctions in marine invertebrates and that of terrestrial tetrapods correlate well with the coincidental anomaly of global and habitat surface temperatures during biotic crises,'. However, it is not accurate that this has not been previously quantitatively evaluated. In particular,

Song et al 2021 (Nature Communications) has also published a quantitative analysis of extinction magnitude and temperature change which appears to show, with a larger, statistical analysis, similar conclusions to those stated here (there is also a relevant response paper McPherson et al. 2022 Results in Engineering). E.g. Song et al 2021, which is omitted from the citations of the submitted MS, already concluded, 'The results show that both the rate and magnitude of temperature change are significantly positively correlated with the extinction rate of marine animals.' There is also a branch of the literature considering specifically the correlations and potential periodicity of extinction and bolide impacts. I believe the author of the current MS needs to explain and adequately justify what it is about their findings that is novel with regard to the recent literature for publication to be considered.

*Author replies for Comment 1*

Thank you for your important comments. For your comment 1, I added results of Song et al 2021 (Nature Communications) in the text (lines 33-34, 43-46, 247-249, 261-265

highlighted by green). I used McPherson et al. 2022 in the text (lines 30, 277-279

marked by green). Song et al 2021 show a good relationship ($R$ = 0.63) between temperature change and marine extinction rate. The novelty of my study is (i) a significant relationship between temperature change and terrestrial tetrapod extinction magnitude (correlation coefficient $R$ = 0.95 for genus and 0.98 for species), (ii) a significant relationship between extinction magnitude and the global and habitat

[marine or terrestrial realm] surface temperature anomalies, (iii) comparison of marine invertebrate and terrestrial tetrapod response for temperature change and explanation of the different extinction magnitudes, (iv) usage of only data having coincidence of mass extinctions and temperature changes in the same outcrop of marine sedimentary rocks resulting in higher relationship ($R$ = 0.92 and 0.95 for genus and 0.88 and 0.95

for species under comparable data for terrestrial tetrapod extinction magnitude)

between temperature change and marine extinction magnitude than that of Song et al

2021. I added these in the manuscript (lines 220-223, 245-247, 324-326, marked by light blue and green). The novelty has already been written in Abstract and

Conclusions.

*Comment 2*

2. Table 1 shows that the submitted study is based on secondary data compiled from the references indicated there, covering a small sample of 7 geological boundaries.

However, it has not been adequately demonstrated that these secondary data are directly comparable. E.g. There are a range of different methods available for calculating extinction magnitudes and it has not been demonstrated that the compiled data use comparable measures e.g. interval lengths, precise choice of numerator and denominator etc. An analogous point also applies to the temperature proxy data.

*Author replies for Comment 2*

I use the conventional method (total number of extinction genera for a mass extinction
interval / total number of genera in a substage just before the extinction) to calculate
genera extinction % of terrestrial tetrapods in all crises studied and marine genera
extinction % of the end-Guadalupian crisis, because those data fit to this method but
not for a new method of Stanley (2016). Marine genera extinction % data of Sepkoski
(1996) and Bambach (2006) correspond to the conventional method. The substage
intervals are more similar to those of Bambach (2006). Therefore, I used those
extinction % data based on the conventional method to compare marine animal
extinction % with terrestrial tetrapod extinction % for the seven biotic crises. I added
these in the manuscript (lines 59-66, 113-117, 151-152, 233-238, 245-247, highlighted
by green). I added Table 3.

*Comment 3*

3. There is apparently no statistical analysis provided to test the presented results or
conclusions. Furthermore, there is a small sample size of 7 geological boundaries
indicated in Table 1, with only 2 events outside the traditional big 5 extinctions. In
contrast, for example Song et al 2021 and Fan et al 2020 (Science) have published
large statistical analyses, of consistent datasets covering complete series of extinction
magnitudes (not hand-selected examples), to test correlations between extinction and
environmental proxies.

*Author replies for Comment 3*

Although Song et al. (2021) analyzed all data of extinctions and sea surface temperature
(SST) changes, there are no confirmation of exact coincidence between extinction rate
and temperature change for minor extinctions. I use only data showing coincidence of
marine extinction horizons and temperature changes in the same outcrop of marine
sedimentary rocks to reach the truth on relationships between extinction magnitude and
surface temperature change in each biotic crisis. Therefore, I analyze the six mass
extinctions and the modern extinction, which coincided with global climate changes.
Explanation on statistical analysis is the same as the reply for comment 2. I added these
in the manuscript (lines 43-48, 245-249, 261-265, marked by green).

*Comment 4*

4. There is currently inadequate consideration of potential effects of sampling bias on
measures such as % extinction. This issue does not appear to be discussed at all despite its considerable importance in this research area. See for example, Alroy (2014
Paleobiology).

*Author replies for Comment 4*

For consideration of potential effects of sampling bias, I separated data of marine taxa
extinction % into three data sets; one is a data group calculated by Sepkoski (1996)
with low extinction values (0–5 %) of G–L and H–A, second one is Bambach (2016)
with the low extinction values, and the third one is Stanley (2016) based on a new
method with the low extinction values, because low extinction values do not change
largely based on different methods (marked by three types of blue circles in Figure 3). I
compared the data based on the conventional methods [Sepkoski (1996) and Bambach
(2016) for marine animals, data calculated from Benton (2013) and Sahney and Benton
(2017) for terrestrial tetrapods] for both marine and terrestrial to get the four
conclusions. Even when I use the other data set based on the new method of marine
animals (incomparable data sets for terrestrial data), the figure shows the same
conclusions. This confirms the conclusions. I added these in the manuscript (lines 59-
66, 114-117, 151-152, 222-225, 245-247 marked by green and light blue).

Words highlighted by light blue, green, and yellow have been revised in the manuscript.
Light blue: for referee #1
Green: mainly for referee #2
Yellow: duration of climate changes and the others
I revised the climate change at the F-F crisis from warming to cooling, because
warming occurred longer term between the two crises, the Lower Kellwasser and the
Upper Kellwasser crises, and shorter-term global cooling episodes separately occurred
in the two crises (lines 167-169, 211-212, Figures 2, 3, Tables 1, 3).

-------------------------------------------------------------------------------------------------
03 May 2022
**Associate Editor decision: Reconsider after major revisions**
by Petr Kuneš
**Comments to the author**:
Thank you for your detailed replies to both reviews. They identified serious issues with
the scientific significance and novelty of the paper as well as the quality of presentation
of the outcomes.
I invite you to undertake a major revision of your manuscript, after which it will be considered again. Please focus especially on the issues raised by reviewer two regarding scientific novelty, presentation of results, statistical evaluation of your data, including sampling bias.

I have revised on them as explained in the above replies.

Kunio Kaiho

---

## Author Response (AR3)

**Author's response for comments of referees and Associate Editor**

***Comments of referee #1***

Throughout the paper, and especially in Section 3.3, you use the term 'correlated', and yet I can see no correlation analysis or test of correlation (e.g. Pearson/ Spearman/

Kendall coefficient of determination). In a sense, such an attempt to fit a straight line would be pointless because the number of points is small, and you are claiming the correlation is with the magnitude of the temperature shift, not its direction, so some are negative, some positive. I guess one could make all temperature shifts positive and then do a line-fit and Pearson r2. But you'd have to factor in reasonable error terms on both estimated temperature anomalies and estimated extinction magnitudes, and these errors might be larger than the 5% you suggest.

But, I'm not sure you should use the word 'correlated' if that has not been tested – just refer to a positive relationship…

*Author replies for comments of referee #1*

Words highlighted by light blue, green, and yellow have been revised in the manuscript marked-up.

Light blue: for referee #1

Green: mainly for referee #2

Yellow: for Associate Editor (major revision), duration of climate changes, and the others

Grey: for Associate Editor (minor revision)

Thank you for your comments.

I added Pearson's correlation coefficient $R$ between marine extinction % and absolute

SST anomaly ($R = 0.92$–$0.95$ for genera) and that between terrestrial extinction % and absolute land temperature anomaly ($R = 0.95$ for genera) marked by light blue. I added

Table 3 to show Pearson's correlation coefficient $R$.

I use "correlated" as "corresponding to". I revised "correlate" to "correspond to" marked by light blue.

I revised "good correlation" to "significant relationship" marked by light blue.

To show difference of extinction % in cooling and warming cases, I revised the following sentence in Abstract and Conclusions (yellow highlighted parts are revised).

The loss of more than 35 % of marine genera and 60 % of marine species loss corresponding to major mass extinctions so called "big five" correlate with a > 7 °C global cooling and a 7–9 °C

global warming for marine animals, and a > 7 °C global cooling and a > ~7 °C global warming for terrestrial tetrapods, accompanied with ± 1 °C error in the temperature anomalies as the global average, although number of terrestrial data is small.

I revised marine genera and species loss % highlighted by yellow in 3.3 because I

added Sepkoski data.

I revised the climate change at the F–F crisis from warming to cooling, because warming occurred longer term between the two crises, the Lower Kellwasser and the

Upper Kellwasser crises, and shorter-term global cooling episodes separately occurred in the two crises (lines 180-183, 225-228, Figures 2, 3, Tables 1, 3).

Minor changes

Line 142: marking the end of the Paleozoic [not Mesozoic]! Done

Line 163: crises = crisis Done

Line 192: O-S; H-A – add to explanations in caption. I revised "O–S" in Figure 3 to end-

O, which is the same as the other figures. In the caption, I added "H–A: Holocene–

Anthropocene." in the caption.

Kunio Kaiho

--------------------------------------------------------------------------------------------------

***Comments of referee #2***

*Comment 1*

1.  The novelty of this study has not been established. The MS says 'relationships between... physical conditions and the magnitude of animal extinctions have not been quantitatively evaluated. My analyses show that the magnitude of major extinctions in marine invertebrates and that of terrestrial tetrapods correlate well with the coincidental anomaly of global and habitat surface temperatures during biotic crises,'. However, it is not accurate that this has not been previously quantitatively evaluated. In particular,

Song et al 2021 (Nature Communications) has also published a quantitative analysis of extinction magnitude and temperature change which appears to show, with a larger, statistical analysis, similar conclusions to those stated here (there is also a relevant response paper McPherson et al. 2022 Results in Engineering). E.g. Song et al 2021, which is omitted from the citations of the submitted MS, already concluded, 'The results show that both the rate and magnitude of temperature change are significantly positively correlated with the extinction rate of marine animals.' There is also a branch of the literature considering specifically the correlations and potential periodicity of extinction and bolide impacts. I believe the author of the current MS needs to explain and adequately justify what it is about their findings that is novel with regard to the recent literature for publication to be considered.

*Author replies for Comment 1*

Words highlighted by light blue, green, and yellow have been revised in the manuscript.

Light blue: for referee #1

Green: mainly for referee #2

Yellow: for Associate Editor (major revision), duration of climate changes, and the others

Grey: for Associate Editor (minor revision)

Thank you for your important comments. For your comment 1, I added results of Song et al 2021 (Nature Communications) and McPherson et al. 2022 in Introduction and

Discussion. Song et al 2021 show a good relationship ($R = 0.63$) between temperature change and marine extinction rate. The novelty of my study is (i) a significant relationship between temperature change and terrestrial tetrapod extinction magnitude (correlation coefficient $R = 0.95$ for genus and 0.98 for species); (ii) a significant relationship between marine and terrestrial extinction magnitude and the global and habitat [marine or terrestrial realm] surface temperature anomalies; (iii) comparison of marine invertebrate and terrestrial tetrapod response for temperature change and explanation of the different extinction magnitudes; (iv) usage of only data having coincidence of mass extinctions and temperature changes in the same outcrop of marine sedimentary rocks resulting in higher relationship ($R = 0.92$ and 0.95 for genus and 0.88 and 0.95 for species under comparable data for terrestrial tetrapod extinction magnitude) between temperature change and marine extinction magnitude than that of

Song et al 2021 ($R = 0.63$), as described in the first paragraph of Discussion. Using these findings lead to the other novelty, which is "The Anthropogenic future extinction magnitude will not reach the major mass extinction magnitude, when the Anthropogenic future extinction magnitude will be parallel to global surface temperature anomaly" which has been added in Abstract and Conclusions. This differs from Song et al 2021.

I added "Although Song et al. (2021) claimed that a temperature increase of 5.2 °C above the pre-industrial level at present rates of increase would likely result in mass extinction comparable to that of the major Phanerozoic events, regardless of other, non-climatic anthropogenic changes that negatively affect animal life; the temperature increase is not 5.2 °C, but 9 °C. The 9 °C

global warming will not appear in the Anthropocene at least till 2500 under the worst scenario (*IPCC, 2013*; IUCN 2021; Tebaldi, et al., 2021). Prediction of the Anthropogenic future extinction magnitude using only surface temperature is difficult, because the causes of the anthropogenic extinction differ from causes of mass extinctions in geologic time. However, I

can predict that the Anthropogenic future extinction magnitude will not reach the major mass extinction magnitude, when the Anthropogenic future extinction magnitude parallelly changes to global surface temperature anomaly." at the end of Discussion.

*Comment 2*

2. Table 1 shows that the submitted study is based on secondary data compiled from the references indicated there, covering a small sample of 7 geological boundaries.

However, it has not been adequately demonstrated that these secondary data are directly comparable. E.g. There are a range of different methods available for calculating extinction magnitudes and it has not been demonstrated that the compiled data use comparable measures e.g. interval lengths, precise choice of numerator and denominator etc. An analogous point also applies to the temperature proxy data.

*Author replies for Comment 2*

I use the conventional method (total number of extinction genera for a mass extinction interval / total number of genera in a substage just before the extinction) to calculate genera extinction % of terrestrial tetrapods in all crises studied and marine genera extinction % of the end-Guadalupian crisis, because those data fit to this method but not for a new method of Stanley (2016). Marine genera extinction % data of Sepkoski (1996) and Bambach (2006) correspond to the conventional method. The substage intervals are more similar to those of Bambach (2006). Therefore, I used those extinction % data based on the conventional method to compare marine animal extinction % with terrestrial tetrapod extinction % for the seven biotic crises. I added these in the manuscript (lines 78-80, 165-166, 248-250, 286-289 highlighted by green).

I added Table 3.

*Comment 3*

3. There is apparently no statistical analysis provided to test the presented results or conclusions. Furthermore, there is a small sample size of 7 geological boundaries indicated in Table 1, with only 2 events outside the traditional big 5 extinctions. In contrast, for example Song et al 2021 and Fan et al 2020 (Science) have published large statistical analyses, of consistent datasets covering complete series of extinction magnitudes (not hand-selected examples), to test correlations between extinction and environmental proxies.

*Author replies for Comment 3*

I added Pearson's correlation coefficient $R$ between marine extinction % and absolute

SST anomaly ($R = 0.92$–$0.95$ for genera) and that between terrestrial extinction % and absolute land temperature anomaly ($R = 0.95$ for genera) marked by light blue. I added

Table 3 to show Pearson's correlation coefficient $R$. These results are shown in

Abstract, Results, Discussion, and Conclusions marked by light blue.

Although Song et al. (2021) analyzed all data of extinctions and sea surface temperature (SST) changes, there are no confirmation of exact coincidence between extinction rate and temperature change for minor extinctions. I use only data showing coincidence of marine extinction horizons and temperature changes in the same outcrop of marine sedimentary rocks to reach the truth on relationships between extinction magnitude and surface temperature change in each biotic crisis. Therefore, I

analyze the six mass extinctions and the modern extinction, which coincided with global climate changes. Explanation on statistical analysis is the same as the reply for comment 2. I added these in the manuscript (lines 36-38, 43-45, 289-294 marked by green and yellow).

*Comment 4*

4. There is currently inadequate consideration of potential effects of sampling bias on measures such as % extinction. This issue does not appear to be discussed at all despite its considerable importance in this research area. See for example, Alroy (2014

Paleobiology).

*Author replies for Comment 4*

For consideration of potential effects of sampling bias, I separated data of marine taxa extinction % into three data sets; one is a data group calculated by Sepkoski (1996)

with low extinction values (0–5 %) of G–L and H–A, second one is Bambach (2016)

with the low extinction values, and the third one is Stanley (2016) based on a new method with the low extinction values, because low extinction values do not change largely based on different methods (marked by three types of blue circles in Figure 3). I

compared the data based on the conventional methods [Sepkoski (1996) and Bambach (2016) for marine animals, data calculated from Benton (2013) and Sahney and Benton
(2017) for terrestrial tetrapods] for both marine and terrestrial to get the conclusions.
Even when I use the other data set based on the new method of marine animals
(incomparable data sets for terrestrial data), the figure shows the same conclusions.
This confirms the conclusions. I added these in the manuscript (lines 76-80, 128-131,
165-166, 236-239, 286-292 marked by green and light blue).

I revised the climate change at the F–F crisis from warming to cooling, because
warming occurred longer term between the two crises, the Lower Kellwasser and the
Upper Kellwasser crises, and shorter-term global cooling episodes separately occurred
in the two crises (lines 180-183, 225-228, Figures 2, 3, Tables 1, 3).

Kunio Kaiho

-------------------------------------------------------------------------------------------------
*25 May 2022*
*Associate Editor decision: Reconsider after major revisions*
*by Petr Kuneš*

**Comments to the author**:
Thank you for performing the major revision and following the reviewers' comments.
After evaluating your revision, I am not entirely satisfied with addressing all the issues.

In particular, I believe that the introduction needs more clarification and justification as
to why your work would bring novel insights into the climate-extinction relationship.
*Author reply*:
Words highlighted by light blue, green, and yellow have been revised in the manuscript.
Light blue: for referee #1
Green: mainly for referee #2
Yellow: for Associate Editor (major revision), duration of climate changes, and the
others
Grey: for Associate Editor (minor revision)

Thank you for your comments. I agree with your comments and added some
words and sentences. The novel insights are clarifying of similarity and difference in
response of terrestrial tetrapods and marine animals for global surface temperature and habitat (land and sea) temperature changes using only biotic crises having
coincidental abrupt surface temperature anomaly (major five mass extinctions and end-
Guadalupian). I added "--- using only biotic crises coinciding with abrupt climate changes, to
access similarity and difference in response of terrestrial and marine animals for global and
habitat (land and sea) temperature anomalies and coincidental environmental changes.
Song et al. (2021) claimed that a temperature increase of 5.2 °C above the pre-industrial
level at present rates of increase would likely result in mass extinction comparable to that of the
major Phanerozoic events, regardless of other, non-climatic anthropogenic changes that
negatively affect animal life. The 5.2 °C is not a global surface temperature anomaly but a sea
surface temperature (SST) anomaly. The global surface temperature anomaly is much higher
than 5.2 °C. Fig. 1d shows the conversion between the global surface temperature anomaly,
land-surface temperature anomaly (global mean), and SST anomaly (global mean) to access
global and habitat (land and sea) temperature anomalies in each biotic crisis. I reached different
conclusions on the surface temperature anomaly and the prediction for the future extinction
magnitude for the conclusions of Song et al. (2021)." in the final part of Introduction. I revised a
conclusion of Song et al. (2021) at the end of the sections 4.1 and 4.2 (lines 294-299, 344-352).
I added "The Anthropogenic future extinction magnitude will not reach the major mass
extinction magnitude, when the extinction magnitude parallelly changes with global surface
temperature anomaly." at the end of Abstract and Conclusions.
It requires a more extended overview of previous studies and their finding, not just
mentioning in one sentence (such as Song et al. 2021), and their fitting into a more
general context, which would be better understandable for the reader (perhaps by
using some of the text you added to the next chapter).
*Author reply*: I added the following sentences in Introduction.
On the modern Earth, an ongoing species extinction occurred mainly on land rather than
the sea (Barnosky et al., 2011). A study on thermal tolerance of modern animals shows a higher
sensitivity of marine animals to warming than terrestrial animals (Pinsky et al., 2019). However,
whether this relationship holds true for ancient animals has not yet clarified. ------ Song et al.
(2021) claimed that a temperature increase of 5.2 °C above the pre-industrial level at present
rates of increase would likely result in mass extinction comparable to that of the major
Phanerozoic events, regardless of other, non-climatic anthropogenic changes that negatively
affect animal life. The 5.2 °C is not a global surface temperature anomaly but a sea surface
temperature (SST) anomaly. The global surface temperature anomaly is much higher than 5.2
°C. Fig. 1d shows the conversion between the global surface temperature anomaly, land-surface
temperature anomaly (global mean), and SST anomaly (global mean) to access global and habitat (land and sea) temperature anomalies in each biotic crisis. I reached different conclusions on the surface temperature anomaly and the prediction for the future extinction magnitude for the conclusions of Song et al. (2021)."

Please explain better why you aimed to clarify the relationship and why it is so important to repeat that! Moreover, the last sentence in the introduction should be better explained concerning the previous content.

*Author reply*: I moved the last sentence to the above paragraph, and added new sentences in the introduction to show why I aimed to clarify the relationship (lines 45-

51). "On the modern Earth, an ongoing species extinction occurred mainly on land rather than the sea (Barnosky et al., 2011). A study on thermal tolerance of modern animals shows a higher sensitivity of marine animals to warming than terrestrial animals (Pinsky et al., 2019). However, whether this relationship holds true for ancient animals has not yet clarified. I aimed to clarify the relationship between the magnitude of biotic crises in not only marine invertebrates but also terrestrial vertebrates (tetrapods) and the global and habitat [marine or terrestrial realm] surface temperature anomalies using only biotic crises coinciding with abrupt climate changes, to access similarity and difference in response of terrestrial and marine animals for global and habitat (land and sea) temperature anomalies and coincidental environmental changes."

I added "The Anthropogenic future extinction magnitude will not reach the major mass extinction magnitude, when the extinction magnitude parallelly changes with global surface temperature anomaly." in Abstract and Conclusions; "However, I can predict that the

Anthropogenic future extinction magnitude will not reach the major mass extinction magnitude, when the Anthropogenic future extinction magnitude parallelly changes to global surface temperature anomaly." at the end of Discussion.

Please, do not mix methods with discussion. I think that all the arguments to support your results should be moved to discussion, e.g., line 63-66.

*Author reply*: I moved the sentences to the second paragraph of discussion 4.1.

Chapter 2.3 - please provide in more detail what kind of analysis did you use to calculate the correlation? Is it Pearson or something else? How did you text the significance? And change it throughout the text.

*Author reply*: I used Pearson (the results are same as those by Correl). I wrote it in

Methods 2.3 and Table 3. The significance of the correlation is very high correlation (0.92-0.95 in marine genera compared with 0.63 in marine genera of Song et al.)

between temperature and extinction magnitude in land and sea. I wrote this in abstract, discussion 4.1, and conclusions.

In the first paragraph of the discussion, you should better highlight the novelty of your
results.
*Author reply*: I exchange the first and second paragraph of 4.1, and revised the
sentences to show novelty of my results [(I)–(IV)] in 4.1.
The other novelty is the additional sentences "Although Song et al. (2021) claimed that a
temperature increase of 5.2 °C above the pre-industrial level at present rates of increase would
likely result in mass extinction comparable to that of the major Phanerozoic events, regardless
of other, non-climatic anthropogenic changes that negatively affect animal life; the temperature
increase is not 5.2 °C, but 9 °C. The 9 °C global warming will not appear in the Anthropocene at
least till 2500 under the worst scenario (IPCC, 2013; IUCN 2021; Tebaldi, et al., 2021).
Prediction of the Anthropogenic future extinction magnitude using only surface temperature is
difficult, because the causes of the anthropogenic extinction differ from causes of mass
extinctions in geologic time. However, I can predict that the Anthropogenic future extinction
magnitude will not reach the major mass extinction magnitude, when the Anthropogenic future
extinction magnitude parallelly changes to global surface temperature anomaly." at the end of
Discussion.
The last sentence reads like a speculation, do you have any better explanation for that
supported by your or other data?
*Author reply*: I revised it to "The correlation coefficient of Song et al. (2021) is much lower ($R$
$= 0.63$ for genus), which is likely due to the low correlation in low extinction rates. It is likely due
to the lack of sensitivity of marine animals for small temperature change or the usage of an
uncertain coincidence with global climate changes." (lines 292–294).
*14 June 2022*
*For Associate Editor (minor revision)*
*Author reply*: I revised the introduction based on the revision of Associate Editor.

Kunio Kaiho